# First-Principles Study of the Reaction between Fluorinated Graphene and Ethylenediamine

**DOI:** 10.3390/molecules24020284

**Published:** 2019-01-14

**Authors:** Jin Tian, Yuhong Chen, Jing Wang, Tingting Liu, Meiling Zhang, Cairong Zhang

**Affiliations:** 1State Key Laboratory of Advanced Processing and Recycling of Non-ferrous Metals, Lanzhou University of Technology, Lanzhou 730050, China; Tianjin10@163.com (J.T.); zhcrxy@lut.cn (C.Z.); 2School of Science, Lanzhou University of Technology, Lanzhou 730050, China; ttlLUT@163.com (T.L.); zhangml_2000@126.com (M.Z.); 3Department of Applied Physics, Nanjing University of Science and Technology, Nanjing 210094, China; mirror1217@163.com

**Keywords:** density functional theory, ethylenediamine, fluorinated graphene, reaction energy barrier

## Abstract

The reaction process between gauche- and trans-structure ethylenediamine (EDA) and fluorinated graphene (CF) was studied based on density functional theory (DFT). Firstly, the reaction between the most stable gauche-structure EDA and CF was discussed. Some of the reaction results were verified in experiment, but the overall reaction energy barrier was higher. Then, the reaction between the trans-structured EDA and CF was simulated, which concluded that CF is reduced in the main reaction channel and HF is generated at the same time. In this reaction process, the reaction energy barrier is as low as 0.81 eV, which indicates that the reaction may occur spontaneously under natural conditions The Mulliken charge population analysis and the calculation of bond energy prove that the NH bond is more stable than CH and that the H atoms in the CH_2_ of trans-structure EDA more easily react with CF.

## 1. Introduction

Graphene is a popular two-dimensional material, where the carbon atoms are bonded by sp2 hybrid orbitals [1,2]. The special structure of graphene means it has excellent electrical conductivity [3], large specific surface area [4], and high tensile strength [5]. Therefore, graphene has attracted great attention from the scientific community. The University of Manchester’s Novoselov first obtained single-layer graphene by simple mechanical stripping in 2004 [6]. However, the large-scale preparation of monolayer graphene is still a challenge. At present, the preparation methods of graphene mainly include the micromechanical exfoliation method [2], epitaxial growth method [7], graphite intercalation method [8], chemical vapor deposition method [9], and oxidation method [10]. The method for preparing graphene by reducing oxide graphene has the characteristics of low cost and high yield, which is suitable for its large-scale production and is favored by scientists [11,12]. Graphene oxide with diamine compounds, specifically EDA, was observed by Che and coworkers in 2010 [13]. However, there are many kinds of oxygen-containing groups in graphene oxide, which necessitates the use of various reducing agents to reduce them. At the same time, the preparation also introduces a variety of impurities which are difficult to remove completely [14]. Moreover, the graphene oxide sheets are prone to agglomeration with the decrease of oxygen-containing groups [15], which affect the excellent physical and chemical properties of graphene itself.

In 1947, by precise control of temperature, Rudorff [16] synthesized a new material with excellent inertness and thermal stability, named CF_0.676_–CF_0.985_ [17]. In recent years, with the research boom in carbon materials, fluorinated graphite has entered researchers’ field of vision with its excellent performance. Due to the electronegativity of the fluorine atom, the interlayer spacing of graphite after fluorination is expanded from 3.35 Å to 7.08 Å, and the interlayer energy is also reduced from 37.6 kJ/mol to 8.4 kJ/mol [18], which indicates that fluorinated graphite is easy to strip into CF [19]. The reaction process for preparing graphene is relatively simple, because the surface of CF only has fluorine atoms [20]. Recently, researchers have used a variety of reducing agents to reduce CF in the experiment, such as hydrazine hydrate [21] and ethylenediamine [22]. Hou and coworkers found that the C/F ratio of CF can be readily tuned by adjusting the reaction temperature with EDA, which leads to defluorination and also substitution of a small amount of fluorine atoms by amino groups [23]. Based on the generalized gradient approximation (GGA) method, He et al. simulated the reaction process between hydrazine and CF [24]. He explained the phenomenon of the experiment and studied its reaction mechanism. However, the reaction energy barrier of the process is generally high. EDA has the characteristics of simple structure, strong reducibility, and rotational isomerism of the molecule [25]. The electron-diffraction data and the quantum chemical calculation indicate that EDA is mainly in the gauche conformation (the N–C–C–N dihedral angle is in the range of 64 ± 4°) [26,27] in the gaseous phase. The trans form of crystalline EDA is obtained through X-ray treatment at −60 °C. [28]. In 2016, Balabaev studied the ethylenediamine configuration in the liquid phase by the molecular dynamics method and found that the molecular structure would be transformed from a gauche to a trans structure at a certain temperature [29].

Therefore, based on density functional theory (DFT), the gauche- and the trans-structure ethylenediamine [30] were selected in this study. The reaction process of different parts of ethylenediamine attacking the surface of the fluorine graphene were simulated to study the experiments’ reaction process of preparing graphene and obtain an optimized reaction path. Moreover, we explain the phenomenon of the experiment and provide a reference for improvement of the experiment.

## 2. Materials and Methods 

Calculations are performed using the CASTEP module in Materials Studio 7.0 software (Accelrys, San Diego, CA, USA), which is based on the density functional theory (DFT) [31], and the DFT evaluation was based on the plane wave expansion. The exchange-associative interaction uses the Perdew-Burke-Ernzerhof (PBE) exchange correlation functional under generalized gradient approximation (GGA), and the ultrasoft pseudopotential (USPP) [32,33] is used to describe the interaction between electrons and ions. All atoms are assumed to be relaxed in the calculation. The convergence criterion of the optimized structure is that the total energy variation is less than 1.0 × 10^−5^ eV/atom, the force per atom is less than 0.03 eV/Å, and the tolerance offset is less than 0.001 Å. Considering the test results and the calculation cost, the cutoff can be selected as 330 eV and the sample of the Brillouin zone K point is 3 × 3 × 1.

Considering the rotational isomerism of ethylenediamine (NH_2_–CH_2_–CH_2_–NH_2_), the most stable gauche configuration and the substable trans configuration of ethylenediamine are the initial configurations. The structural optimization of the two structures of ethylenediamine molecules can obtain the energy of gauche structures close to that of trans structures. The energy of gauche structures and trans structures are −980.446 eV and −980.395 eV, respectively, and the structures are shown in Figure 1a,b. The gauche-type N–C–C–N has a dihedral angle of 61.9°, which is agreement with the N–C–C–N dihedral angle range of 64 ± 4° [26] reported by Yokozeki. The computation results show that the gauche structure of ethylenediamine has lower energy and a more stable structure.

Ao et al. [34] reported the CF chair structures, which have lower energy and are more stable. We have performed a parameter optimization by ourselves, and the optimization results contrast with those of Ao and coworkers. A 4 × 4 × 1 supercell was built, in which the carbon-fluorine atomic ratio was 1:1. The optimized CF bond is 1.384 Å (experimental value is 1.380 Å [35]). In order to avoid interaction between the layers, the vacuum layer is set to 20 Å in the direction of the vertical carbon layer surface. The optimized chair-shaped fluorinated graphene C_32_F_32_ structures are shown in Figure 1c,d.

In the calculation of the transition state, the ethylenediamine molecule was placed at a position that is 5 Å away from the surface of CF. Optimized structures are regarded as the initial state (IS), and the product after the reaction is taken as the final state (FS). The linear synchronous transit and the quadratic synchronous transit are combined to search for the transition state, and the main reaction channel is determined by the reaction energy barrier. When the reaction energy barrier is lower than 0.91 eV, the reaction may spontaneously proceed under standard conditions [36].

## 3. Results and Discussion

Three reaction paths of the ethylenediamine molecule and fluorinated graphene are mainly studied: one in which the F on the CF surface is attacked by the H in the NH_2_, the second in which the H atom in CH_2_ attacks the F atom on the surface of the fluorinated graphene, and the third in which the H atoms in the NH_2_ and CH_2_ in the ethylenediamine simultaneously attack the F atom on the fluorinated graphene.

### 3.1. Reaction of Gauche-Structure EDA and CF 

Yokozeki [26] reported that the structure of EDA is mainly based on the gauche type. Firstly, the reaction of CF and EDA in the gauche structure was studied. The reaction R1 involves the attack of the F atom by the H atom in the NH_2_. After the reaction, the F atom is reduced and an N–F bond is formed that has a bond length of 1.427 Å. The structural formula of the EDA molecule becomes H_2_NH_2_C–CHNH_2_F, as shown in Figure 2a. In reaction R2, the H atom in the CH_2_ of EDA attacks the surface of the CF. After the reaction, one H atom on the C atom of EDA and one F atom on fluorinated graphene form an H–F bond. Meanwhile, the structural formula of EDA becomes H_2_NH_2_C–CHNH_2_, as shown in Figure 2b. There are two reaction paths, R3 and R4, when the surface of the CF is attacked simultaneously with H atoms in NH_2_ and CH_2_. The reaction results are shown in Figure 2c,d, respectively. In reaction R3, the EDA molecule is broken into NH_2_ and CH_2_CH_2_NH_2_. The NH_2_ is decomposed into NH and H on the surface of the fluorinated graphene. The H atom replaces one F atom, and the NH replaces two F atoms. The N atom in the NH forms a C–N bond with two C atoms, one F atom forms an H–F bond with the H atom on the carbon in CH_2_CH_2_NH_2_, and the other two F atoms are bonded to the H atom and the F atom on CH_2_CHNH_2_, respectively, to become FH_2_NHC–CH_2_F. After the end of the reaction, the CF surface is distorted and N atoms are present, which is in agreement with the experimental results of Hou and Ma [23,37]. The C–N bond formed in reaction R4 is also cleaved. During the reaction, one F atom is reduced from the CF bonds with the N atom of NH_2_ to become NH_2_F and H_2_C–CH_2_NH_2_.

Wagner [38] showed that the gauche structure of EDA becomes more stable due to intramolecular hydrogen bonds. Table 1 (R1–R4) lists the reaction formula, the reaction energy barrier corresponding to the transition state, the product energy barrier, and the reaction energy of the entire process of the gauche-structure EDA and CF. The results show that the reaction of the gauche structure of EDA and CF needs to overcome a certain reaction energy barrier, which means that the reaction needs to obtain a certain amount of energy from the outside. The most prone reaction is R1, which requires overcoming the reaction energy barrier of 4.357 eV to occur, and absorbing external energy of 1.319 eV throughout the process. Reaction R2 needs to cross the reaction energy barrier of 5.339 eV, and the energy of 0.907 eV is released in the reaction, meaning it is an exothermic reaction. The reaction barrier of reaction R3 is the largest (16.397 eV), but three F atoms on the surface of the fluorinated graphene are replaced and reduced. Reaction R4 needs to cross the reaction barrier of 12.938 eV to occur. It can be seen that the reaction of EDA and the gauche-structure CF needs to overcome the higher reaction barrier and to consume a large amount of external energy, as shown in Figure 3a. Therefore, the reaction of the trans-structured EDA and CF was further studied. 

### 3.2. Reaction of Trans-Structure EDA and CF

In reaction R5, the H atom in NH_2_ of the trans EDA attacks the surface of the CF, and the result is shown in Figure 2e: the replacement reaction between the H atom in the NH_2_ and the F atom on the surface of the fluorinated graphene results in the minor change of the molecular structure of the post-trans-structure ethylenediamine. The molecular formula changes to H_2_NH_2_C–CH_2_NHF, where the bond length of the NF and the CH bonds are 1.472 Å and 1.106 Å, respectively. In reaction R6, the H atom in CH_2_ of the trans EDA attacks the surface of the CF. The reaction results that are shown in Figure 2f are similar to those of R2: one F atom on the surface of the CF bond reacts with an H atom in CH_2_ to form HF. By analyzing the Mulliken charge population of R6 products, we found that the charge population of the H_2_NCH_2_–CHNH_2_ is 0.96 e, the charge population of the HF is 0, and the charge population of the C_32_F_31_ is −0.95 e. H_2_NCH_2_–CHNH_2_ is the carbon-centered radical, and the original molecular structure undergoes a large rotational deformation. 

In reactions R7 and R8, the H atom on the CH_2_ and NH_2_ simultaneously attack the surface of the CF. The results of reaction R7 are shown in Figure 2g: H atoms on the NH_2_ and CH_2_ in the ethylenediamine molecule are respectively substituted with the two F atoms on the fluorinated graphene and the molecular formula of EDA is changed to H_2_NH_2_C–CHFNHF, in which the bond lengths of NF and CF are 1.456 Å and 1.417 Å, respectively. The two H atoms also replace the positions of F atom on the surface of fluorinated graphene, and the bond lengths of CH are 1.99 Å and 1.098 Å, respectively. The reduction reaction occurred during reaction R8. The result is shown in Figure 2h: the C–C bond in the EDA is broken into two NH_2_CH_2_ molecules. The C atom in the one of the NH_2_CH_2_ molecules is bonded with an F atom in the surface of the CF, and the C–N single bond becomes a double bond in the other NH_2_CH_2_ molecule.

Table 1 shows the reaction formulas R5–R8 of the trans-structured EDA and CF, the reaction energy barrier, product energy barrier, and reaction energy of the corresponding transition state. In four reactions, R5 is an endothermic reaction, R6–R8 are exothermic reactions, and a more stable product is obtained after the exothermic reaction. Reaction R5 needs to overcome the reaction energy barrier of 2.358 eV and absorbs 0.220 eV of energy from the outside during the whole reaction. The reaction energy barrier of R6 is 0.810 eV; after completion of the reaction, 0.815 eV energy is released, and the reaction energy barrier satisfies the condition that it is possible to spontaneously proceed under standard conditions. Reactions R7 and R8 need to cross the reaction energy barrier of 5.731 eV and 10.830 eV, respectively. The reaction energy barrier is obviously increased, as shown in Figure 3b. According to the reaction energy barrier, the trans-structured ethylenediamine more easily reacts with fluorinated graphene. Among them, the R6 reaction is more stable and the advantage of it is more obvious.

In summary, reaction R6 is the main reaction channel of the first step of reaction of EDA and CF, and the product is used to investigate the subsequent reaction. H atoms in the CH and NH_2_ of the product (H_2_NH_2_C–CHNH_2_) of reaction R6 attack the surface of the CF. The F atom in the CF is easily reduced due to the bond unsaturation in the reactant H_2_NH_2_C–CHNH_2_ molecule. It is bonded with a carbon atom to form a stable product H_2_NFHC–CH_2_NH_2_, where the C–F bond is 1.446 Å long, as shown in Figure 2i (denoted as reaction R6-1). In reaction R6-2, an H atom on NH_2_ attacks the surface of CF, H forms a bond with F atoms shed on CF, and the H_2_NHC-CH_2_NH_2_ molecule becomes NH_2_CH_2_–CHNH. The reaction energy barriers of R6-1 and R6-2 are 1.899 eV and 1.945 eV, respectively, which are higher than the energy barrier of the reduction reaction R6 for the first step. The reaction is only carried out under appropriate external conditions. The energy barrier of reaction R6 is 0.810 eV, which is lower than the condition of the lowest reaction energy barrier that may be reacted under natural conditions as mentioned in the transition state theory [36]. In the subsequent reaction, the reaction difficulty increases with the increase of the reaction energy barrier. The actual reaction solution contains an excess of EDA in the experiment, so the reaction mainly takes the R6 reaction pathway. Reaction R6 reduces the F atom on the surface of the CF, and after the reaction occurs a plurality of times, the CF is reduced to graphene.

Further analysis found that the gauche structure of EDA is more stable than the trans structure due to intramolecular hydrogen bonding, while the trans structure has higher reactivity. The reaction energy barrier of trans-structure EDA and CF is significantly lower than the reaction energy barrier of the gauche structure, and the reaction of the trans structure is more likely to occur. The reaction process of the trans-structure EDA with CF is more likely to occur when the surface of the CF is attacked by the H atom in CH_2_. The Mulliken charge population can analyze the bond strength and charge transfer. The Mulliken charge population analysis of trans EDA found that the charge population of the H atoms in NH_2_ is 0.41–0.42 e, the charge population of the N atoms is −0.96 e, the charge population of the H atoms in CH_2_ is 0.26–0.27 e, and the charge population of the C atoms is −0.40 e. The Mulliken charge population analysis of gauche EDA found that the charge population of the H atoms in NH_2_ is 0.40–0.42 e, the charge population of the N atoms is −0.96 e, the charge population of the H atoms in CH_2_ is 0.27–0.29 e, and the charge population of the C atoms is −0.42 e. As a result, it can be seen that the charge transfer between the N atom and H atom is much larger than that between the C atom and H atom. The charge transfer between the C atom and H atom of the gauche conformer is bigger than that of the trans conformer.

We calculated the C−H bond energy $E_{b}\left(C-H\right)$ and the N–H bond energy $E_{b}\left(N-H\right)$ of the gauche conformer and the trans conformer. The binding energy $E_{b}\left(C-H\right)$ and $E_{b}\left(N-H\right)$ are given as:
$$E_{b}\left(C-H\right)=-E_{EDA}+E_{NH2CH2-CHNH2}+E_{H}\;E_{b}\left(N-H\right)=-E_{EDA}+E_{NHCH2-CH2NH2}+E_{H}$$
where $E_{EDA}$, $E_{NH2CH2-CHNH2}$, $E_{NHCH2-CH2NH2}$ and $E_{H}$ represent the total energy of EDA, the energy of NH_2_CH_2_–CHNH_2_, the energy of NHCH_2_–CH_2_NH_2_, and the energy of the H atom, respectively. Average of $E_{b}\left(C-H\right)$ in the trans structure is 5.708 eV, and in the gauche conformer, it is 5.775 eV. Moreover, the average of $E_{b}\left(N-H\right)$ in the trans structure is 5.852 eV, and in the gauche conformer, it is 5.881 eV. Therefore, $E_{b}\left(N-H\right)$ is more stable than $E_{b}\left(C-H\right)$, and the $E_{b}\left(C-H\right)$ of the gauche conformer is more stable than that of the trans conformer; that is, the trans conformer is more reactive than the gauche conformer. The calculated results of bond energy are in agreement with those of Mulliken charge population analysis. As a result, the H atom in the trans structure ethylenediamine more easily reacts with the fluorinated graphene.

The density of states (DOS) reflects the number of states of the unit energy, which is important in further understanding the reaction between EDA and CF. The interactions between C and F atoms on graphene fluoride before and after the reaction were analyzed. Partial densities of states (PDOS) of C and F are shown in Figure 4a. Due to the existence of the C–F bond, the 2p orbit of the C atom and the 2p orbit of the F atom have overlapping peaks at −9.2 eV before reaction. After the reaction of R6, there is no overlapping peak between the 2p orbit of the C atom and the 2p orbit of the F atom, which means the C–F bond is broken. This result is consistent with the reaction phenomena. The EDA and the CF DOS diagram before and after reaction are analyzed further, as shown in Figure 4b,c. After the R6 reaction, the DOS diagram has been changed obviously: the CF and the EDA DOS diagram have moved to the left. It means that the system becomes more stable when the reaction is completed. The product of R6, C_32_F_31_, has conductivity according to the DOS of Figure 4b. With the decrease of F atoms in CF, the conductivity of the system is increased, which is consist with the experimental result of Lee [39].

## 4. Conclusions

In summary, based on the density functional theory, the reaction process between the EDA that has the most stable gauche structure and the substable trans structure, respectively, and CF was studied. The following conclusions were obtained:

(1) The energy barrier of the main reaction channel R6 is 0.810 eV, which can be spontaneously carried out under natural environmental conditions. In the subsequent reaction, the reaction difficulty increases with the increase of the reaction energy barrier. When there is an excess of EDA, the R6 reaction is dominant. After the R6 reaction occurs multiple times, the fluorinated graphene is reduced to graphene.

(2) The EDA of the gauche structure is more stable than that of the trans structure, while the trans structure has higher reactivity.

(3) The charge transfer between N atoms and H atoms in the gauche- and the trans-structured EDA is much larger than that between C atoms and H atoms, and the bond energy $E_{b}\left(N-H\right)$ in the gauche- and trans-structure EDA is much larger than that of the bond energy $E_{b}\left(C-H\right)$. So, the NH bond is more stable than the CH bond and is less prone to fracture. The charge transfer between C atoms and H atoms in the trans structure is smaller than that in the gauche structure, and the bond energy $E_{b}\left(C-H\right)$ in the trans structure is smaller than that in the gauche structure as well. The result is that the H atoms in the CH_2_ of trans-structure EDA is more prone to react with CF.

## Figures and Tables

**Figure 1 molecules-24-00284-f001:**
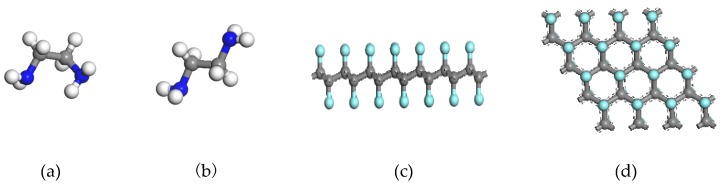
(**a**) Gauche-structure ethylenediamine (EDA); (**b**) trans-structure EDA; (**c**) fluorinated graphene (CF) chair structure: front view; (**d**) CF chair structure: top view. Gray, blue, white, and cyan colors represent C, N, H, and F atoms, respectively.

**Figure 2 molecules-24-00284-f002:**
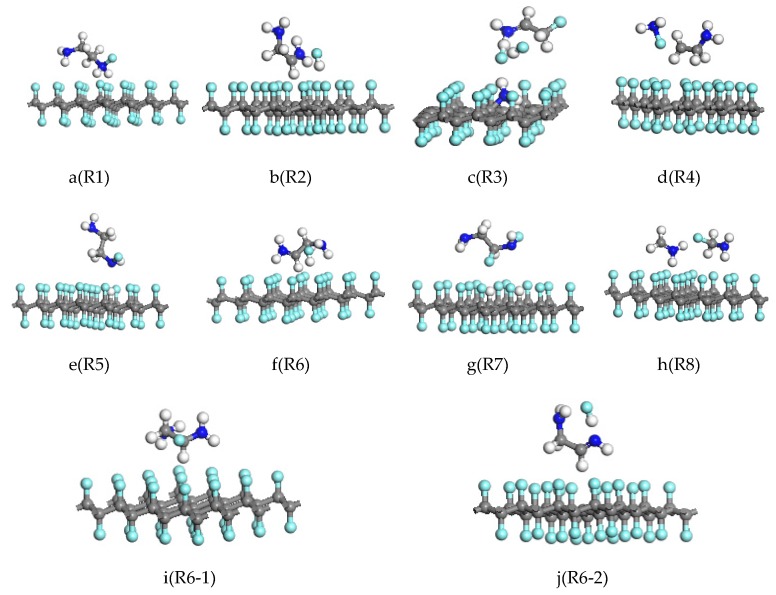
The structures of products of the reaction of the gauche structure (**a**–**d**) and the trans structure (**e**–**h**) of ethylenediamine with fluorinated graphene. (**i**,**j**) represent subsequent reactions of R6. Gray, blue, white, and cyan colors represent C, N, H, and F atoms, respectively.

**Figure 3 molecules-24-00284-f003:**
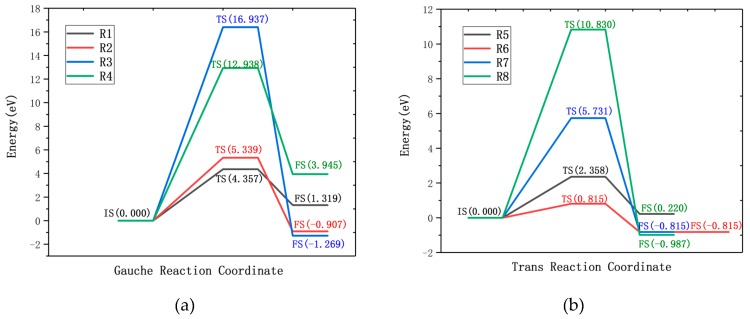
Reaction energy profiles of the R1–R4 reaction paths (**a**) and the R5–R8 reaction paths (**b**), Where the IS, TS and FS represent the initial state, transition state and final state, respectively.

**Figure 4 molecules-24-00284-f004:**
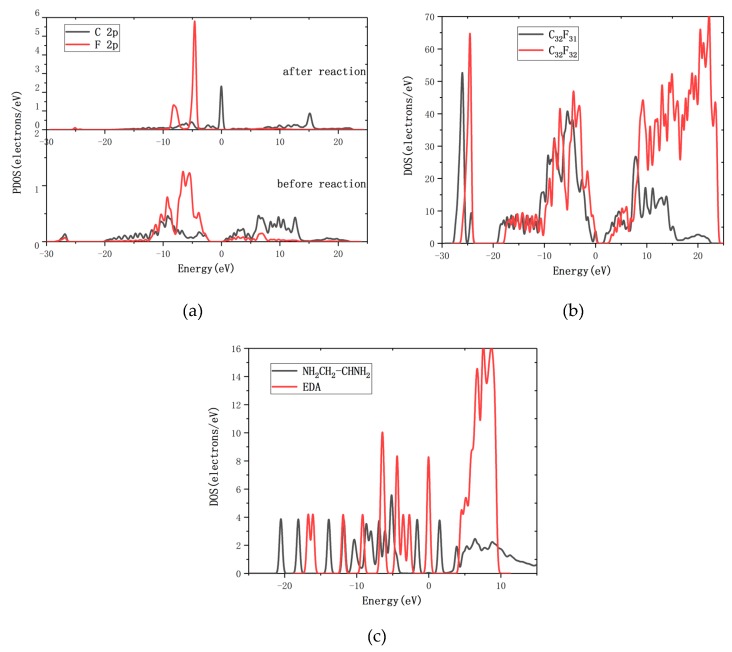
(**a**) The partial densities of states (PDOS) of C and F atoms on graphene fluoride before and after the reaction; (**b**) the CF DOS diagram before and after reaction; (**c**) the EDA DOS diagram before and after reaction.

**Table 1 molecules-24-00284-t001:** Reaction process and reaction parameters of EDA and CF.

	Reactions	Barrier from Reactant (eV)	Barrier from Product (eV)	Energy of Reaction (eV)
R1	C_32_F_32_ + H_2_NH_2_C–CH_2_NH_2_ → C_32_F_31_ + H_2_NH_2_C–CHFNH_2_F	4.357	3.038	1.319
R2	C_32_F_32_ + H_2_NH_2_C–CH_2_NH_2_ → C_32_F_31_ + H_2_NH_2_C–CHNH_2_ + HF	5.339	6.246	−0.907
R3	C_32_F_32_ + H_2_NH_2_C–CH_2_NH_2_ → C_32_F_29_NH_2_ + HF + FH_2_NHC–CH_2_F	16.397	17.666	−1.269
R4	C_32_F_32_ + H_2_NH_2_C–CH_2_NH_2_ → C_32_F_31_ + NH_2_F + H_2_C–CH_2_NH_2_	12.938	8.984	3.945
R5	C_32_F_32_ + H_2_NH_2_C–CH_2_NH_2_ → C_32_F_31_H + H_2_NH_2_C–CH_2_NHF	2.358	2.178	0.220
R6	C_32_F_32_ + H_2_NH_2_C–CH_2_NH_2_ → C_32_F_31_ + H_2_NHC–CH_2_NH_2_ + HF	0.810	1.625	−0.815
R7	C_32_F_32_ + H_2_NH_2_C–CH_2_NH → C_32_F_30_H_2_ + H_2_N_2_HC–CHFNHF	5.731	6.546	−0.815
R8	C_32_F_32_ + H_2_NH_2_C–CH_2_NH_2_ → C_32_F_31_ + CH_2_NH_2_ + CH_2_FNH_2_	10.830	11.817	−0.987
R6-1	C_32_F_32_ + H_2_NHC–CH_2_NH_2_ → C_32_F_31_ + H_2_NFHC–CH_2_NH_2_	1.899	2.956	−1.057
R6-2	C_32_F_32_ + H_2_NHC–CH_2_NH_2_ → C_32_F_31_ + HNHC–CH_2_NH_2_ + HF	1.945	2.865	−0.920

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
