# Peer review of "First-Principles Study of the Reaction between Fluorinated Graphene and Ethylenediamine"

_molecules, 2019, doi:10.3390/molecules24020284_

Round 1

Reviewer 1 Report

This manuscript contains some novel findings on the reaction mechanism of two ethylenediamine comformers with fluorinated graphene. The subject of this study is appropriate for publication in Molecules. However, several revisions described below should be considered prior to acceptance for publication.

(1) It has been described that the C-N bond of trans-EDA converts to double bond (C=N?) through the reaction path R6. The product of the R6 is H2NCH2–CHNH2 and HF; in this case, the bond between CH and NH2 is not a double bond because the valence of N in NH2 is insufficient. This product may be carbon-centered radical rather than unsaturated molecule. Structure of the product and related reaction pathways should be reconsidered. 

(2) In the last part of the discussion, the bond strength of  N-H and C-H was evaluated using the calculation results of the Mulliken charge population. However, the bond strength should be evaluated on the basis of the bond energy (sum of the energies of dissociated products – the energy of reactant; in this case, the total energy of H2NCH2CHNH2 or H2NCH2CH2NH + H – the energy of EDA). In addition, this comparison does not explain why the trans conformer is more reactive than the gauche conformer.

(3) It is concluded that the trans conformer is more reactive than the gauche conformer, but the last sentence of the discussion is inconsistent with this conclution (line 201–202: … H atom in the gauche structure ethylenediamine more easily reacts …).

(4) The value of the reaction barrier of R1 described in the main text (line 125: 4.375 eV) is inconsistent with that listed in Table 1.

(5) Atom number contained in the chemical formula should be written as a subscript. 

Reviewer 2 Report

The authors have studied the interaction between ethylenediamine and fluorinated graphene. In the manuscript, we find some interesting results that deserve to be published. But, in its actual form, I cannot recommend the publication of this manuscript. First of all, English have to be improved. On the other hand, from a scientific point of view, the interest of this project is not properly motivated. Relative to the methodology, It is not clear at all if the authors have performed a parameters optimization by themselves or they have just taken the values from Ao et al. Furthermore, at the beginning of the Materials and Methods section the state: ‘ultrasoft pseudopential (USPP) [30] is used to describe the interaction between electrons’. The statement is very vague. It is difficult to believe that they really used USPP to describe all electrons. I found the analysis of the data rather poor, for example, I am missing some analysis of the density of state. It would be also interesting to show the energy profile of the reaction paths, showing clearly the transition states. The authors should also show the Muliken analysis for gauche EDA. Finally, as the author propose reaction R6 as a way to reduce fluorinated graphene to graphene, they should discuss the effect of EDA-EDA interaction, would reaction R6 remain unaltered in the presence of other EDA molecules?

Round 2

Reviewer 1 Report

The revised manuscript has been sufficiently improved. 

Reviewer 2 Report

The authors have significantly improved the quality of their manuscript. They have properly addressed most of my comments. However, there are still some issues that have to be addressed by the authors. The interest of the project is still not properly motivated in the introduction. It would be also very interesting to show the energy profile of the reaction paths, showing the transition states.
